



# Multi-event assessment of typhoon-triggered landslide susceptibility in the Philippines

Joshua N. Jones[1,2], Georgina L. Bennett[2], Claudia. Abancó[3], Mark A. M. Matera[4], Fibor J. Tan[4]

[1]AECOM, East Wing Plumer House, Plymouth, PL6 5DH, United Kingdom
[2]College of Life and Environmental Sciences, University of Exeter, Exeter, EX4 4RJ, United Kingdom
[3]Faculty of Earth Sciences, University of Barcelona, 08028 Barcelona, Spain
[4] School of Civil, Environmental and Geological Engineering, Mapúa University, Manila, Philippines

*Correspondence to:* Joshua N. Jones (joshua.jones1@aecom.com)

**Abstract.** There is a clear and pressing need to improve and update landslide susceptibility models across the Philippines. This is challenging, as landslides in this region are frequently triggered by temporally and spatially disparate typhoon events, and it remains unclear whether such spatially and/or temporally distinct typhoon events cause similar landslide responses, i.e., whether the landslide susceptibility for one typhoon event is similar for another. Here, we use logistic regression techniques (implemented alongside a LASSO (Least Absolute Shrinkage and Selection Operator) for independent variable selection) to develop four landslide susceptibility models based on three typhoon-triggered landslide inventories. These inventories are of landslides triggered by the 2009 Typhoon Parma (local name Typhoon Pepeng), the 2018 Typhoon Mangkhut (local name Typhoon Ompong), and the 2019 Typhoon Kammuri (local name Typhoon Tisoy). The 2009 and 2018 inventories were mapped across the same 150 km2 region of Itogon in the Benguet Province, whilst the 2019 event was mapped across a 490 km2 region of Abuan in the Isabela Province. The four susceptibility models produced are for the 2009, 2018 and 2019 inventories separately, and for the 2009 and 2018 inventories combined. By comparing the susceptibility model outputs across all four models, we are then able to assess the similarity in landslide response between the different typhoon events. Furthermore, using AUROC validation and 30% of the landslide inventories saved for independent testing, we quantify the degree to which susceptibility models derived from one event can forecast/hindcast the landslides triggered by the other events. The logistic regression approach produced susceptibility models with 65 – 82% accuracy, with the 2009, 2018, and combined 2009-2018 models being considerably more accurate (78 – 82%) than the 2019 model (65%). Furthermore, we find that the three typhoon events caused quite different landslide responses. Most notably, landslides in Itogon triggered by the 2018 and 2009 typhoons were heavily distributed across E/SE/S-facing slopes and at slope angles >30o, whereas landslides in Abuan





triggered by the 2019 typhoon occurred across all aspects and slope angles. Finally, the AUROC validation shows that using a susceptibility model for one typhoon event to forecast/hindcast another leads to a 6 – 10%

reduction in model accuracy compared to the accuracy obtained when modelling and validating each event separately. However, using a susceptibility model for two combined typhoon events (2009 + 2018) to forecast/hindcast each typhoon event separately led to just a 1 – 3% reduction in model accuracy. This suggests that combined multi-event typhoon triggered landslide susceptibility models will be more accurate and reliable for the forecasting of future typhoon-triggered landslides.

**1.0 Introduction**

In the Philippines, landslide occurrences and hazards are high (Kirschbaum *et al.* 2015; Lin *et al.* 2017; Abancó *et al.* 2021), with hydrological hazards and associated landslides causing thousands of fatalities and millions of pesos in damage every year. Indeed, across Southeast Asia, approximately 46% of all rainfall triggered landslides occur in the Philippines, of which 42% are triggered by typhoons (Froude & Petley

2018). However, despite the pervasiveness of landslides in the Philippines, high quality country-wide typhoon-triggered landslide susceptibility maps are lacking, thus representing a major resource gap in efforts aimed at better managing and mitigating future landslide hazard across the country. For example, whilst statistical landslide susceptibility studies have been undertaken in the Philippines (e.g. Oh & Lee 2011; Nolasco-Javier & Kumar 2019, 2020), these remain geographically limited, and insufficient for use in

planning purposes. Indeed, as illustrated by Fig. 1, the susceptibility maps currently held by the Philippines Mines and Geosciences Bureau (MGB) uses a heuristic approach, and so only shows only very broad categorisations of landslide susceptibility.

One of the major challenges in developing improved typhoon-triggered landslide susceptibility models across

the Philippines is uncertainty around the spatial and temporal heterogeneity in the occurrence of landslide-triggering typhoons. The problem lies in the fact that it is currently unclear whether spatially and temporally distinct typhoons trigger landslides with similar distributions and susceptibilities. I.e., whether typhoon triggered landslide susceptibility in the Philippines is spatially and/or temporally dependent. This problem is not unique to the Philippines, with studies from other regions showing how landslide susceptibility cannot be

assumed to be spatially and/or temporally dependent (e.g. Jones *et al.,* 2021). Knowing this is important, as if landslide susceptibility is spatially and/or temporally dependent, then it will not be appropriate to use landslide susceptibility models developed from one typhoon event to forecast landslides triggered by future typhoon events or by typhoon events in other regions.

*Figure 1. Current landslide susceptibility map of the Itogon region held by the Philippines Mines and Geology Bureau (Mines and Geosciences Bureau (MGB), 2018 ).*



In the literature, there is growing evidence to suggest that rainfall-triggered landslide spatial distributions and susceptibility are indeed spatially and temporally dependent. For example, recent research by Jones *et al.* (2021) shows how landslides triggered by different monsoon seasons and cloud outburst storms in Nepal

have distinctly different spatial distributions and susceptibility, with several other papers also highlighting how landslide susceptibility is commonly both spatially and/or temporally dependent (e.g. Gorsevski *et al.* 2006; Meusburger & Alewell 2009; Lombardo *et al.* 2020; Ozturk *et al.* 2021). However, as stated above, it remains unclear whether typhoon-triggered landslides in the Philippines are also spatially and temporally dependent. This is a problem as rainfall in the Philippines is predicted to increase by 22 – 32% between 2006

– 2035 in the Benguet province (Nolasco-Javier *et al.* 2015), with extreme rainfall projected to increase in provinces such as Luzon (PAGASA, 2022). Therefore, rainfall triggered landslides are likely to become more frequent, and thus it will become increasingly important to be able to understand, forecast and mitigate this hazard.

The overall aim of this paper is to use data from multiple typhoon events to assess typhoon-triggered landslide susceptibility in the Philippines. The specific objectives are as follows: First, to use Binary Logistic Regression (BLR) techniques to develop four landslide susceptibility models across two regions (Itogon and Abuan; Fig. 2) using data from three typhoon events; the 2009 Typhoon Parma and 2018 Typhoon Mangkhut that occurred in Itogon, and the 2019 Typhoon Kammuri that occurred in Abuan. The four models we

developed are for each typhoon event separately, plus for the 2009 and 2018 Itogon events combined. The second objective is to assess the similarities and differences between the susceptibility results obtained from each model. Finally, the third objective is to use Area Under the Receiver Operator Curve (AUROC) validation to quantify whether the models developed for the 2018, 2009, and 2018+2009 models can be used to accurately classify (or forecast/hindcast) the landslides triggered by the other individual typhoon event.

I.e., to assess whether time-independent modelling of typhoon-triggered landslides in the Philippines is appropriate. This objective essentially operates under the hypothesis that if typhoon triggered landslides are time dependent, then in a given region there will be a reduced model accuracy when using a model developed from one typhoon event to classify another. As well as allowing the development of improved and updated landslide susceptibility maps for two regions of the Philippines, completion of these aims will provide

important wider insight into the spatial and temporal dependence of landslide susceptibility modelling.

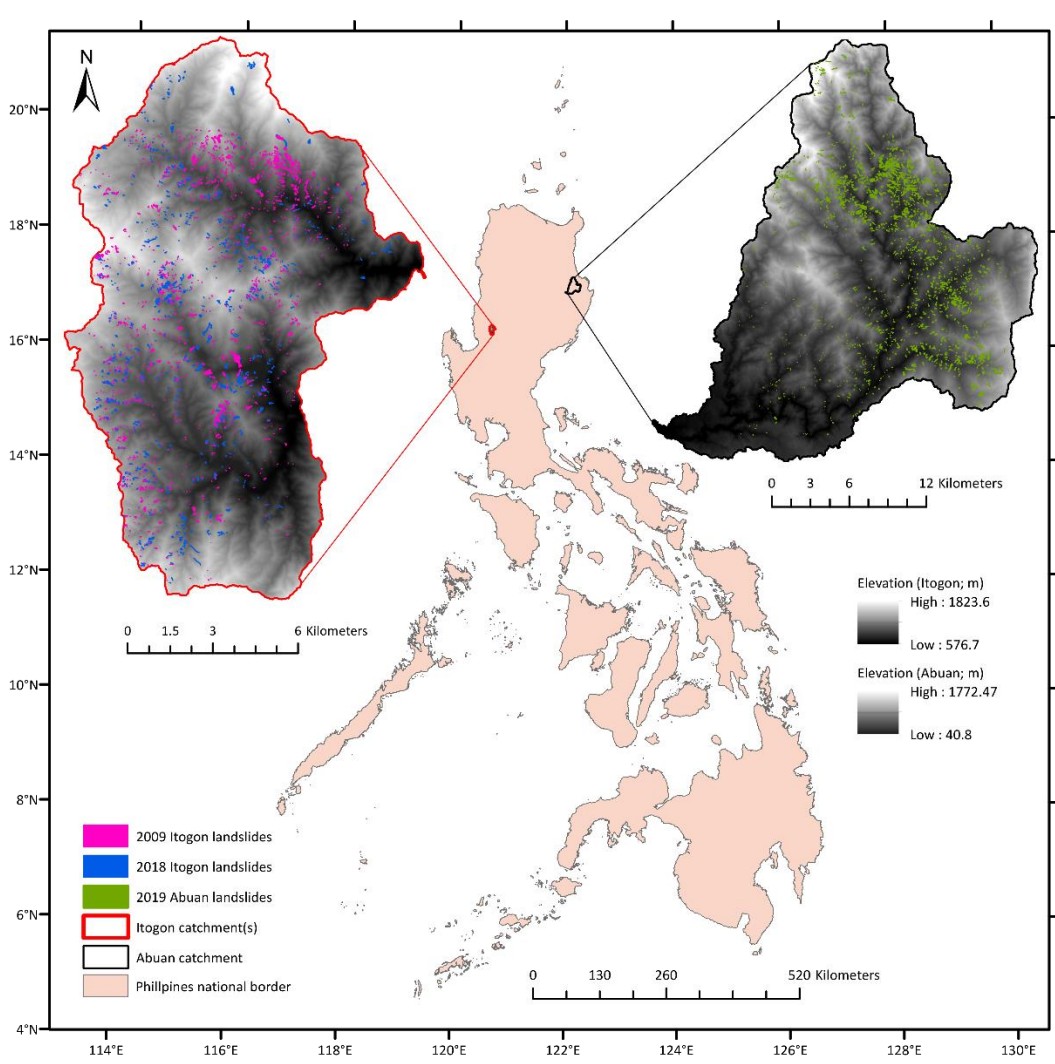

**Figure 2.** *Locations of the two study regions, including the three landslide inventories used throughout this paper.*

## 2.0 Regional setting

The climate in the Philippines is controlled by a variety of interacting systems including the south east monsoon, summer typhoons/cyclones, El Nino and La Nina cycles, and the Inter Tropical Convergence Zone (ITCZ) (Nolasco-Javier *et al.* 2015). The interplay between these systems typically leads to drier conditions



from November to April and wetter conditions from May to October. As stated above, the focus of this paper is on typhoon triggered landslides. As such, the specific sub-regions selected for this study are a group of catchments in the Itogon municipality, Benguet province, and a catchment of Abuan, in the Ilagan municipality, Isabela province (Fig 2). The Itogon region is located along the southern portion of the Cordillera Central Mountains. These catchments all drain into the Agno River, which flows broadly N-S along the eastern side of the study region. This region is located to the north-east of Baguio City, which has a population of ~350,000 people. The Abuan catchment forms the north-east part of the Pinacanauan de Ilagan catchment, which is a major tributary of the Cagayan River. The catchment is estimated to have a population of ~14,000 people and supports large areas of agriculture (Balderama *et al.* 2016). These regions were selected for this study as both have experienced particularly significant landslide-triggered typhoon events over the past few decades. Most notably, the 2009 Parma (known locally as Pepeng) and 2018 Mangkhut (known locally as Ompong) typhoons in Itogon, and the 2019 Kammuri (known locally as Tisoy) typhoon in Abuan. The following paragraphs describe the key characteristics and known landslide information of each of these three typhoon events, before outlining the geological and geomorphological setting of the Itogon and Abuan subregions.

**2.1 Typhoon Parma (Pepeng)**

Typhoon Parma formed on the 27[th] September 2009 and dissipated on the 14[th] October 2009. The main impacts of this typhoon occurred in northern Luzon, particularly across the Itogon region, between the 3[rd] and 9[th] of October, when the total rainfall reached over 1868 mm (Nolasco-Javier *et al.* 2015) and wind speeds of 121 to 150 mph near the centre (NDCC, 2009). The impacts of this typhoon were severe, with some 500 reported fatalities, over $635 million in damages, and at least 60 damaging landslide occurrences (Nolasco-Javier *et al.* 2015; Liou & Pandey 2020). The severity of these impacts was in part due to the simultaneous occurrence of typhoon Melor. The interaction between Parma and Melor led to a phenomenon called the Fujiwhara effect, whereby typhoon Melor caused typhoon Parma to slow, rotate, and loop such that it actually made landfall over northern Luzon three times (Shimokawa *et al.* 2009; Nolasco-Javier *et al.* 2015; Liou & Pandey 2020). Furthermore, these typhoons occurred following several months of El Nino-induced higher than average rainfall, including storm Koppu in September (Yumul *et al.* 2013). This resulted in high antecedent rainfall prior to the start of the typhoon season, which likely enhanced the triggering of landslides (Nolasco-Javier *et al.* 2015). In terms of specific landslide impacts, at least 60 damaging landslides have been reported (National Disaster Coordinating Council 2009). Limited (rapid response) landslide assessment for this event has been conducted in the Baguio region, with the predominant observed landslide types being





slides, debris flows and earth flows, and a tentative minimum triggering rainfall threshold of 70 mm in 24
hours (Nolasco-Javier *et al.* 2015).

### 2.2 Typhoon Mangkhut (Ompong)

Typhoon Mangkhut formed on September 6th 2018, made landfall on the Philippines between the 12th – 15th
of September, and dissipated on September 17th. In terms of rainfall, the highest recorded total precipitation
was 794 mm at the Baguio City PAGASA (Philippine Atmospheric, Geophysical and Astronomical Services
Administration) weather station (Abancó *et al.* 2021). With maximum winds of 121 - 150 mph when it made
landfall (PAGASA, 2018), the impacts of this typhoon were severe, with extensive reported damage to
buildings and homes, loss of power, and over 100 reported fatalities (Sassa 2018; Niu *et al.* 2020).
Furthermore, Mangkhut is known to have triggered thousands of landslides, including one large complex
failure that caused 94 casualties alone (Abancó *et al.* 2021; Kim *et al.* 2021). Recent research by Abancó *et*
*al.* (2021) presents a detailed inventory of over 1,100 landslides triggered by Mangkhut, most of which
occurred across a region of Itogon affected by 360 mm of rainfall over a 44 hour period, according to
satellite-based GPM IMERG rainfall records. These landslides were of all different types, though dominated
by shallow translational landslides and mud and debris flows, with many of the landslides exhibiting complex
behaviour whereby they initiated as shallow slides before transitioning into flows. Most of these landslides
occurred on grassland or wooded east- to southeast-facing slopes underlain by superficial clays (Abancó *et*
*al.* 2021). Further details of the mapping methodology and characteristics of this inventory are provided in
section 3.0.

### 2.3 Typhoon Kammuri (Tisoy)

Typhoon Kammuri formed on the 24th November 2019, made landfall on the 2nd December, and dissipated on
the 6th December 2019. When Kammuri made landfall, it was a category 4 storm with wind speeds of up to
70 - 89 mph (Sevieri & Galasso 2020; PAGASA, 2019). Between the 2nd and 4th of December this event
reportedly damaged or destroyed over 561,000 buildings, caused at least 17 fatalities, and led to economic
losses in excess of $116 million (LeComte 2020; Sevieri & Galasso 2020). In terms of landslides, initial
reports from aid groups suggested that landslide-induced damage to roads and other infrastructure was
widespread (NDRRMC 2019; IFRC 2020). However, there appears to have been no subsequent mapping or
assessment of the landslides triggered by this event.



### 2.4 Geological and geomorphological setting

The Itogon catchments are dominated by a bedrock geology of Cretaceous and Tertiary quartz diorite and andesite/basalt (DENR-MGB, 1995, 2000), with the remainder of the catchment underlain by Quaternary

sandstone, claystone, and conglomerates. The bedrock is typically overlain by superficial deposits of clays, silty/sandy loams, and mountain soils, with a landcover dominated by shrub/grass land and open forest. Geomorphologically, the hillslopes across the Itogon catchment have a mean elevation of 1140 m, mean and maximum hillslope angles of 28º and 71º, an equal distribution of hillslope aspects, and predominantly (60%) concave morphologies. The Abuan catchment is almost 100% characterised by Cretaceous and Tertiary

metamorphosed basic intermediate flows and/or pyroclastics and metamorphosed andesites and basalts (DENR-MGB, 1991a, 1991b, 1976). These are overlain by superficial deposits of mountain soil and a vegetation dominated by open forest. Geomorphologically, the region is dominated by steep uplands and rugged hills, with a lower mean elevation than Itogon of 560 m, similar hillslope angle (mean of 23º and max of 73º) and aspect distributions, but hillslopes that are dominated (60%) by convex morphologies.

### 180 3.0 Data: landslide inventories and predisposing factors

As outlined in subsequent sections, BLR landslide susceptibility modelling requires data on both landslide and predisposing factors (e.g. geology, soil, landcover, topography). The following sections outline the key datasets used throughout this paper, including the mapping procedures and key characteristics associated with each landslide inventory and descriptions of all predisposing factor datasets (topographical, geological, land

use etc.).

### 3.1 Landslide inventories

This paper uses three landslide inventories (Fig. 2), each associated with one of the typhoons in 2018, 2009 and 2019. The 2018 Mangkhut inventory is a slightly clipped version of the inventory presented by Abancó *et al.* (2021), whilst the 2009 Parma and 2019 Kammuri inventories are presented here for the first time.


The employed mapping procedure was the same for all three inventories. In each case, the landslides were initially mapped by one or two mappers and then independently reviewed, checked, and amended as appropriate by a different mapper. All landslides were mapped manually via the visual inspection of pre- and post- typhoon imagery. The 2009 Parma inventory was predominantly mapped GoogleEarth imagery dated

31/12/2003 for pre-event images, and 31/12/2009 for post-event images. The 2018 Mangkhut inventory was





mapped using a combination of 0.5 m WorldView-2, 10 m Sentinel-2, and 3 m Planet Labs imagery, with pre-typhoon images dated between 18/02/2018 and 06/09/2019, and post images dated between 19/09/2018 and 02/03/2019. Finally, the 2019 Kammuri inventory was predominantly mapped using 3 - 5 m resolution Planet Labs imagery, with pre-typhoon images dated between 02/10/2019 and 28/10/2019, and post-typhoon

images dated between 07/01/2020 and 31/03/2020. For all inventories, the limited availability of cloud-free imagery resulted in gaps of several months between the pre- and post-typhoon imagery used. As rainfall triggered landslides in the Philippines are so common, it is therefore possible that some of the landslides visible in the imagery actually occurred before or after the respective typhoon, either due to other rainfall events that occurred within the imagery window, or due to human activity such as construction or mining.

This issue was unavoidable, but mitigated where possible by cross-checking each inventory with other imagery (e.g. available GoogleEarth images) and by qualitative comparisons with local reports and field surveys carried out by the Philippines Mines and Geology Bureau (MGB) (Abancó *et al.* 2021).

For all inventories, landslides were delineated as polygons that included the scar, deposition, and runout

zones of each event. Care was taken to avoid landslide amalgamation, e.g., combining multiple overlapping or contiguous landslides into one unionised polygon (Marc & Hovius 2015), and to avoid the erroneous mapping of non-landslide features such as anthropogenic cut-and-fill or road-tip associated mass-wasting and processes such as channel bank erosion. In total, 1912 landslides were in the 2009 Parma inventory, 956 in the 2018 Mangkhut inventory, and 1964 in the 2019 Kammuri inventory. In each case, to estimate landslide

types, the Aspect Ratios (ARs) of all landslides were calculated. ARs give the ratio between the length (longest) and width (shortest) axis of each landslide. Landslides with AR values between 1 and 2 are more isometric, and thus more likely to be slumps or slides, whereas landslides with AR vales > 4 are more likely to be long runout flow-type landslides. For these inventories, 30%, 21% and 35% of the landslides in the Parma, Mangkhut, and Kammuri inventories respectively had AR values < 2, with 17%, 28% and 13% > 4.

Unfortunately, due to recent travel restrictions it was not possible to ground truth these inventories and their estimated characteristics in the field.

**3.2 Predisposing factor datasets**

For both study regions, all topographic data were derived from 5 m resolution Digital Elevation Models (DEMs) that were obtained in 2013 using IfSAR techniques and provided by the Philippines Department of

Environment and Natural Resources -National Mapping and Resource Information Authority (DENR-NAMRIA). Topographic datasets including slope, aspect and curvature were derived from the DEM using





the Spatial Analysis toolbox in ArcGIS. Distance to streams was also derived from the DEMs using the "STREAMobj" function in the Matlab TopoToolBox (Schwanghart & Scherler 2014). Finally, datasets on bedrock geology, soil cover and land-use were obtained from the following sources:

- Itogon Geology Map: Provided by the Department of Environment and Natural Resources-Mines and Geosciences Bureau (DENR-MGB). This included the geological maps of the Baguio City Quadrangle, Sheet 3169 III (1995), and the Sison Quadrangle, Sheet 3168 IV (2000).

- Abuan Geology Map: Provided by the Department of Environment and Natural Resources-Mines and Geosciences Bureau (DENR-MGB). This included the geological map of the Ilagan
Quadrangle, Sheet 3371 II (1991), the Tumauini Quadrangle, Sheet 3371 I (1991) and the Municipality of Ilagan, Isabela, 1:90K Scale (1976).

- Soil cover (both regions): Provided by the Department of Agriculture-Bureau of Soils and Water Management (DA-BSWM) (Carating 2013).

- Land cover (both regions): Department of Environment and Natural Resources-National Mapping
and Resource Information Authority (DENR-NAMRIA). Data from 2015.

**4.0 Methods**

The first objective of this section is to use Binary Logistic Regression (BLR) techniques to develop four typhoon-triggered landslide susceptibility models and maps for two regions of the Philippines using landslide data from three different typhoon events. The third objective is to use Area Under the Receiver Operator
Curve (AUROC) validation to quantify whether each model can be used to accurately forecast/hindcast independent landslides triggered by each other typhoon event. The following sections will therefore describe the relevant BLR and AUROC methodologies used throughout this paper.

**4.1 Binary Logistic Regression (BLR) modelling**

BLR models can be understood as classification algorithms that are used to classify the binary outcome (0 or
1) of a dependent (or response) variable (e.g., landslide absence / presence in a given grid cell) given a set of independent predictor variables (e.g., landslide predisposing factors such as elevation, geology etc.). The key BLR equation to be used in this paper is given in equation 1, which essentially describes how the probability of landsliding ($\pi$; the dependent variable) is linked to a model intercept ($\beta_0$) and combinations of regression coefficients ($\beta_i$) and independent variables ($X_i$):


$$\pi = \frac{1}{1+10^{(\beta_0 + \beta_1 x_1 + \beta_i x_i)}}$$   **Equation 1**

BLR models are commonly used within the literature to assess landslide susceptibility (Pourghasemi *et al.* 2018; Reichenbach *et al.* 2018). As such, the mathematics behind equation 1 are well described elsewhere
(e.g. Appendix A of Lombardo & Mai 2018) and will not be re-described here. In this paper, we use the glmnet package (Hastie & Qian 2014) in the statistical software, R, to develop BLR models that are implemented alongside a LASSO (Least Absolute Shrinkage and Selection Operator) (e.g., Lombardo & Mai 2018; Jones *et al.* 2021) for variable selection.

The LASSO is an algorithm that automatically determines which independent predictor variables are most important for classifying the response of the dependent variable. Full mathematical descriptions of the LASSO can be found in Friedman *et al.* (2010), Hastie & Qian (2014) and Lombardo & Mai (2018) so will not be repeated here. However, in effect, the LASSO works by cycling through different combinations of increasingly more independent variables by systematically setting different independent variables to zero
until it converges on a user-defined optimal solution (Friedman *et al.* 2010). In this case, the optimal solution was to maximise the model success as defined by the AUROC (Area Under the Receiver Operator Curve; see section 4.2), i.e., the success at that model in classifying the data used to train the model. The advantage of this methodology is that it provides objective information on which combinations of independent variables are having the most dominant influences on landslide occurrence, as well as the usual information on
independent variable regression coefficients.

Before running the glmnet BLR models, data sets of combined landslide and predisposing factor data were obtained using the following steps:
1. Divide each study region into 5x5 m grids ($2.4 \times 10^7$ cells in Itogon, $3.9 \times 10^7$ cells in Abuan)
2. Take each landslide inventory and convert the polygons to "highest points" at the assumed triggering location of each landslide. Then, combine the 2018 and 2009 high points so that we now had four inventories of high points. One each for 2018, 2009, 2019 and 2018+2009.
3. Using each of the four inventories, we then assign each cell in the relevant study region a value of 1 if it observed a landslide high point (landslide presence cell), and a value of zero if not
(landslide absence cell).
4. For each region, we then use the datasets described in section 3.2 to also assign each cell in each study region a value for each predisposing factor of interest. In total, there were nine





predisposing factors, of which three were categorical (geology, soil type and landuse) and the rest were continuous.

5. For each case, we then extract 50 random balanced training sub-datasets, where each subset includes a random selection of 70% all of the landslide presence data plus an equal number of randomly selected landslide absence cells. Note, 50 sub-datasets were used for each model to get an appreciation of error and uncertainty within each model.

6. Finally, 50 random balanced testing sub-datasets were also extracted for each case, where the
testing subsets included the 30% of landslides not selected for the training sets plus an equal number of randomly selected landslide absence cells. These testing sets were used for the model AUROC validation, see section 4.2.

Once the 50 subsets per model had been obtained, there were also several processing steps that had to be
completed before running the models:

7. To ensure that the final regression coefficients assigned to the different independent variables were objectively comparable, all continuous variables were scaled using zero-mean unit variance (e.g., Lombardo & Mai 2018).

8. As the models were to include several continuous independent variables, it was possible that
some variables would actually be collinear. This is potentially problematic, as inter-variable collinearity can make regression models unstable and inaccurate (Zuur *et al.* 2010). Consequently, we tested for collinearity between all continuous variables in all 200 sub-datasets (50 per model) using the Variance Inflation Factor (VIF) functions of Zuur *et al.* (2010). VIFs are a commonly used approach to quantifying collinearity. VIFs were calculated
for every continuous variable of interest, where a VIF value $> 5$ suggests that the associated variable is collinear with at least one other variable in the model and so should be removed. However, in this case, no variables had VIFs $> 2$, suggesting that none of the independent variables were collinear.

With these data sub-setting and processing steps completed, the 50 datasets per case were run through the glmnet model. The resulting intercept values, LASSO selection percentages, and associated regression coefficients were then averaged for each case based on the 50 respective model-runs. These averaged values were then inputted into Equation 1 alongside the relevant predisposing factor datasets (the $x_i$ parts of equation 1) and applied across the entirety of the respective study regions to obtain the final landslide
susceptibility maps for each case. This was done using the ArcGIS raster calculator.




### 4.2 AUROC validation

The Area Under the Receiver Operator Curve (AUROC) is a commonly used method to assess the accuracy and validity of landslide susceptibility maps (Pourghasemi *et al.* 2018; Reichenbach *et al.* 2018). In this paper, AUROC methods are used to assess the initial accuracy of each susceptibility model, as well as to

investigate how well one model can classify independent landslide data from the same or other typhoon events. A Receiver Operator Curve (ROC) is a probability curve obtained by plotting the True Positive Rate (TPR), or sensitivity, against the False Positive Rate (FPR), or 1 – specificity. Where in this case the TPR is the proportion of landslide presence cells correctly classified as landslide presence cells by a model, and the FPR is the proportion of landslide absence cells incorrectly classified as landslide presence cells by a model.

The area under the ROC (the AUROC value) indicates the accuracy with which a given binary model was able to correctly classify the observed classes (in this case the landslide presences and absences). An AUROC value of 1 means that a model was 100% accurate, whilst an AUROC value of 0.5 indicates that a model has zero classification capacity (i.e., is no better than a random guess). An AUROC value < 0.5 indicates that a model is actively inverting the classification (i.e., classifying landslide presences as absences

and vice versa), whilst values of 0.7 – 0.75 are generally taken to represent a good model, and values > 0.8 a very good model (e.g., Marjanović 2013; Vakhshoori & Zare 2018; Jones *et al.* 2021). In this paper, all AUROC values were calculated using 10-fold cross-validation, whereby 100 AUROC validations between the 50 models developed for a given inventory and the 50 independent testing sets from a given inventory were used to calculate an average AUROC value and associated standard deviation.

### 5.0 Results

### 5.1 BLR modelling

As outlined in the methods, four BLR models were developed. Three for the Itogon region based on the 2018 inventory, 2009 inventory, and combined 2018+2009 inventory, and one for the Abuan region based on the 2019 inventory. The key outputs from the BLR models are the average intercepts, average regression

coefficients, LASSO selection percentages, and initial model accuracies (as determined by AUROC analysis), as well as the resulting landslide susceptibility maps.

Table 1 summarises the average intercepts, regression coefficients, LASSO selection percentages and initial model accuracies of each model. The regression coefficients and LASSO selection percentages essentially

describe the influences of each independent factor on the model. A larger magnitude coefficient and higher


| Independent Factor | Itogon 2018 | | Itogon 2009 | | Itogon 2018+2009 | | Abuan 2019 | |
|---|---|---|---|---|---|---|---|---|
| Intercept | -0.1101 | | -0.0149 | | -0.1411 | | -5.0378 | |
| AUROC | 0.83 +/- 0.009 | | 0.80 +/- 0.007 | | 0.78 +/- 0.006 | | 0.68 +/- 0.008 | |
| | Mean Regression Coefficient | LASSO Selection % | Mean Regression Coefficient | LASSO Selection % | Mean Regression Coefficient | LASSO Selection % | Mean Regression Coefficient | LASSO Selection % |
| Distance to channel | **0.073** | **94** | **0.138** | **100** | **0.119** | **100** | **0.033** | **66** |
| Elevation | -0.058 | 58 | -0.088 | 74 | -0.076 | 94 | -0.284 | 100 |
| Planform curvature | **0.555** | **100** | **0.167** | **100** | **0.330** | **100** | **0.385** | **100** |
| Profile curvature | -0.358 | 100 | -0.201 | 100 | -0.253 | 100 | -0.130 | 98 |
| Slope | **0.360** | **100** | **0.433** | **100** | **0.427** | **100** | -0.003 | 28 |
| N | -0.995 | 100 | -2.888 | 100 | -1.478 | 100 | -0.025 | 70 |
| NE | - | 0 | -0.860 | 100 | -0.455 | 100 | 0.152 | 12 |
| E | **0.553** | **100** | **0.146** | **50** | **0.474** | **100** | **0.034** | **74** |
| SE | **0.617** | **100** | **0.520** | **100** | **0.774** | **100** | 0.050 | 12 |
| S | **0.231** | **74** | **0.355** | **100** | **0.584** | **100** | 0.007 | 34 |
| SW | -1.021 | 100 | 0.097 | 6 | - | 0 | -0.037 | 46 |
| W | -1.255 | 100 | -0.931 | 100 | -0.974 | 100 | -0.040 | 52 |
| NW | -1.917 | 100 | -2.872 | 100 | -2.346 | 100 | -0.033 | 46 |
| | | | | | | | | |
| Alluvium | - | 0 | - | 0 | - | 0 | - | 0 |
| Andesite/Basalt | -0.194 | 90 | 0.075 | 24 | 0.027 | 4 | **1.329** | **100** |
| Dacite | **0.138** | **52** | -0.368 | 62 | -0.312 | 84 | - | 0 |
| Limestone | - | 0 | - | 0 | - | 0 | -1.047 | 98 |
| Quartz diorite | **0.189** | **68** | 0.050 | 20 | **0.089** | **94** | -0.900 | 2 |
| Schist | - | 0 | - | 0 | - | 0 | - | 0 |
| Sand/claystone, conglomerate | 0.156 | 4 | -0.400 | 100 | -0.233 | 96 | - | 0 |
| Ultramafic | - | 0 | - | 0 | - | 0 | - | 0 |
| | | | | | | | | |
| Beach sand | - | 0 | - | 0 | - | 0 | - | 0 |
| Clay | **0.416** | **94** | 0.023 | 6 | **0.136** | **62** | - | 0 |
| Clay loam | 0.015 | 2 | 0.252 | 48 | 0.098 | 34 | - | 0 |
| Gravel loam | -1.102 | 32 | -1.005 | 34 | **0.612** | **70** | - | 0 |
| Hydrosol | - | 0 | - | 0 | - | 0 | - | 0 |
| Loam | - | 0 | - | 0 | - | 0 | - | 0 |
| Mountain soil | -1.925 | 100 | -2.726 | 100 | -3.069 | 100 | - | 0 |
| Sandy clay | - | 0 | - | 0 | - | 0 | - | 0 |
| Silty sandy loam | -1.147 | 100 | -0.066 | 24 | -0.284 | 92 | - | 0 |
| | | | | | | | | |
| Barren | -0.419 | 54 | -1.015 | 76 | -1.167 | 96 | - | 0 |
| Builtup | -0.383 | 90 | -0.138 | 44 | -0.289 | 94 | - | 0 |
| Closed forest | - | 0 | - | 0 | - | 0 | - | 0 |
| Cropland | -0.479 | 54 | -0.247 | 46 | -0.340 | 88 | -0.454 | 6 |
| Mangrove | - | 0 | - | 0 | - | 0 | - | 0 |
| Open forest | 0.136 | 10 | -0.014 | 4 | - | 0 | **3.805** | **100** |
| Shrub/grassland | **0.105** | **50** | **0.339** | **100** | **0.159** | **100** | -0.187 | 8 |
| Water | -1.208 | 38 | -1.088 | 50 | -1.689 | 90 | -0.477 | 12 |

**Table 1.** *Summary of the BLR-LASSO results obtained for each of the four developed models. All numbers are the mean of the 50-runs undertaken for each model. The AUROC value is the accuracy of the model at classifying the landslide data used to train the model.*






selection percentage suggests that a factor is more dominant in controlling landslide occurrence, with a negative regression coefficient meaning a factor is making landsliding less likely, and a positive regression coefficient meaning a factor is making landsliding more likely. The regression coefficients and LASSO

selections highlight several similarities and differences between the landslides triggered by each typhoon event. In terms of similarities, the landslides triggered by all three typhoon events have similar relationships with distance to channels, planform curvature, profile curvature and elevation, all of which were consistently selected factors by the LASSO. Increasing distance to channels and planform curvature is found to make landslides slightly more likely to occur, whilst increasing elevation and profile curvature makes landslides

less likely to occur. It is unclear why increasing distance from channels makes landslides more likely, but this could be related to rainfall distributions or slopes being steeper further from channels. Future work should consider this result in more detail. However, it should be noted that the regression coefficient for elevation during 2019 was notably larger (-0.28 compared to -0.06 and -0.09), and thus more dominant, than it was in 2018 and 2009.


For the other topographical factors included in the modelling, it is evident that their impacts on landsliding are similar in 2018 and 2009, but different in 2019. For example, in 2018 and 2009, higher slope angles made landslides more likely (as expected). However, in 2019 slope angle was not found to be a dominant control on landsliding, with less than 30% of the model runs selecting slope as an important factor, and those that did

assigning it a coefficient value near zero (i.e., it has little to no effect on landsliding). Though landslides were still unlikely at slopes lower than 10 degrees. Similarly, the control of aspect on landsliding is relatively consistent between 2018 and 2009. For example, in both cases, north, northwest, and west facing slopes were consistently selected by the LASSO and found to make landslides less likely, whilst east, southeast, and south facing slopes were also consistently selected and found to make landslides more likely. However, in

2019, the regression coefficients for all aspects were near zero, suggesting that aspect was not exerting a dominant control on landsliding in 2019. This result is highlighted in Fig. 3, which graphically displays the aspect regression coefficient and LASSO selection results presented in Table 1.





**Figure 3.** *Graphical display of the aspect regression coefficients presented in Table 1. Red line shows the "zero line" for the regression coefficients, black markers show the regression coefficients and associated +/- 1SDs, and the bars show the selection percentages.*



For the categorical model factors (geology, soil type and land cover), the influences on landsliding are less consistent across each typhoon event. One of the few similarities is that mountain soil and barren land both had > 50% LASS0 selection rates and negative regression coefficients in 2018 and 2009, though neither factor were selected by the LASSO in 2019. The only other similarity between events was that shrub/grassland had high selection rates and positive regression coefficients in 2019 and 2009, though again

was not selected in 2019. In fact, the only categorical factors to be selected consistently by the LASSO in 2019 were andesite/basalt, and open forest, which had strong positive coefficients, and limestone, which had a strong negative coefficient.  Other notable results are that in 2018 the clay unit and silty/sandy loam units had 100% LASSO selection rates, with the former making landslides more likely and the latter making them less likely, whilst in 2019, the sandstone/claystone/conglomerate unit had 100% selection rates and made

landslides less likely.

**5.2 Landslide susceptibility maps**

Fig. 4 shows the final landslide susceptibility maps resulting from the model parameters presented in Table 1. As shown in table N, these maps had initial accuracies (i.e., the accuracies at classifying the data used to train them) of 68 - 83%. However, to properly validate the accuracies of these maps it was necessary to use

AUROC analysis to assess how well each model could classify independent testing data from each of the respective events. The resulting AUROC curves and AUROC values are shown in Fig. 5, which show that the independently tested accuracies of each model are 65 - 82%. As highlighted in section 4.2, these AUROC values suggest that the three Itogon models are good to very good (accuracies of 78 – 82%), but that the Abuan model is poor (accuracy of 65%).

**5.3 Cross-model AUROC validation**

To assess the temporal dependency of typhoon-triggered landslides, we used cross-model AUROC validation to investigate how well one model could predict independent landslide testing data from the other typhoon event(s). I.e., we assessed the AUROC for the 2018+2009 model's ability to classify independent landslide data from the 2018, 2009 and 2019 typhoon events (Fig. 6)., the AUROC for the 2018 model's ability to

classify the 2009 and 2019 events (Fig. 7a - b), and the AUROC for the 2009 model's ability to classify the 2018 and 2019 events (Fig. 7c - d). Note, as the 2019 model had low initial accuracy even when trying to classify the landslides used to train it, we did not cross-validate this model against the 2018 or 2009 landslide data.



**Figure 4.** *The final susceptibility maps produced from: a) the 2018 landslide inventory, b) the 2009 landslide inventory, c) the combined 2018 and 2009 inventories, and d) the 2019 inventory. Where a - c) correspond to the Itogon region, and d) to the Abuan region.*




*Figure 5. Receiver Operator Curves (ROCs) showing the success of a) the 2018 model classifying the independent 2018 testing data, b) the 2009 model classifying the independent 2009 testing data, c) the 2019 model classifying the independent 2019 testing data, and d) the 2018+2009 model classifying the independent 2018+2009 testing data. In each case the reference ROC for a "random" model with 50% accuracy is shown, as is the mean Area Under the Receiver Operator Curve (AUROC) value for all ROCs*


**Figure 6.** *Receiver Operator Curves (ROCs) showing the success of the 2018+2009 model at classifying independent testing data from a) 2018, b) 2009, c) 2019. In each case the reference ROC for a "random" model with 50% accuracy is shown, as is the mean Area Under the Receiver Operator Curve (AUROC) value*

*for all ROCs.*

**Figure 7.** *Receiver Operator Curves (ROCs) showing the success of a) the 2018 model classifying independent testing data from 2009, b) the 2018 model classifying independent testing data from 2019, c) the 2009 model classifying independent testing data from 2018, and d) the 2009 model classifying independent testing data from 2019. In each case the reference ROC for a "random" model with 50% accuracy is shown, as is the mean Area Under the Receiver Operator Curve (AUROC) value for all ROCs.*





This shows that the 2018 and 2009 are 6 - 10% worse at classifying the other individual typhoon events

compared to the models developed specifically for those events (e.g., Fig. 5). Conversely, the 2019 model is
32-33% worse at classifying the 2018 and 2009 models compared to the models developed specifically for
those events, with AUROC values > 0.6 in both cases. Finally, the 2018+2009 model is 8% worse at
classifying the 2019 event compared to the model developed specifically for 2019, but only 1 - 3% worse at
classifying the 2009 and 2018 events compared to the models developed specifically for those events.

**6.0 Discussion**

**6.1 Landslide susceptibility modelling**

The first and second aims of this paper were to develop new and updated susceptibility models and maps for
two regions of the Philippines using three typhoon-triggered landslide events, and to assess the similarities
and differences between the susceptibility results of the different models. As shown by the AUROC values,

in the Itogon region this aim has been met, with three models developed with good to very good AUROC
values of 78 – 82%. Furthermore, these maps can be compared to the existing susceptibility map (Fig. 1) held
by the MGB for this region. By comparing the existing and new maps, it is evident that these maps agree on
the broad scale susceptibility classification, i.e., that much of the Itogon region has high to very high
landslide susceptibilities. However, the new maps have much more slope-scale detail, whereby it is possible

to distinguish differential susceptibility across different topographical characteristics of the landscape.
Indeed, as also highlighted by the regression coefficients (Table 1), it is clear that typhoon-triggered landslide
in susceptibility in Itogon is highest at E/SE/S aspects, slope angles of 30 – 40º, convex curvatures, clay soil,
and shrub/grass land use, but lower at W/NW/N aspects, slope angles < 30º, concave curvatures and
mountain soil. These results are unsurprising given the observations of Abancó *et al.* (2021) who found

similar topographical and geological relationships between the landslides associated with the 2018 Mangkhut
typhoon. As such, due to the increased detail and information provided, the new Itogon maps should be much
more useful for the purposes of hazard zonation and management than the existing MGB map.

Conversely, the model developed for the Abuan region is significantly less accurate, with AUROC values of

just 65- 68%. Unfortunately, the MGB do not currently hold landslide susceptibility maps for this region. So,
at the most fundamental level, on the assumption that some information is better than no information, the
Abuan model produced here is a step forward. However, it is clear that the model developed for Abuan is not


as accurate as the models developed for Itogon. This raises several questions that will be considered in the following paragraphs.


First, why is the Abuan model so much less accurate than those developed for Itogon? One explanation is that the model inaccuracies are due to biases and incompleteness in the 2019 Kammuri landslide inventory. Work by Steger *et al.* (2016, 2017, 2021) outlines how inconsistent, biased, incomplete, or otherwise inaccurate landslide inventories can lead to the development of statistical susceptibility models with incorrect

or unfeasible regression coefficients. It is difficult to quantify whether this issue is relevant in this case. However, given that the 2019 inventory was mapped, checked, and reviewed by four different people, and mapping was conducted using the same methodology as was used for the 2018 and 2009 inventories, it seems unlikely that mapping error can explain the significant model inaccuracies.  Furthermore, whilst the inventories were based on imageries with slightly varying resolutions, the numbers of landslides mapped for

the Abuan case suggests that this did not affect the completeness or accuracy of the landslides in the inventory.

Another explanation for the reduced accuracy of the 2019 model could lie in the input data. For the Itogon region, the categorical inputs for geology, soil, and landcover included multiple classes, with landslides

occurring within several of these classes (Fig. 8a - c). However, in the Abuan region, these inputs were of a lower resolution, with near-homogenous classes across the entire region, and almost all landslides just occurring in one class of each variable (Fig. 8a - c). Indeed, in Abuan there is only one soil type, "mountain-type soil", so it is impossible for the regression model to use soil as an input type (hence why there are no regression coefficient values for soils in Abuan in Table 1). Consequently, there was far less input data

available for the BLR model to use in the Abuan case compared to the Itogon case. Indeed, the Itogon region has several soils, which will all likely have different geotechnical characteristics (e.g. shear resistance etc.), which may aid the model in differentiating landslide susceptibility classes. Similarly, it is evident that the landslides included in the 2019 inventory had significantly different distributions to the landslides associated with the 2018 and 2009 typhoons. Whilst the 2018 and 2009 landslides were preferentially distributed at

certain aspects and slope angles (e.g. Fig. 8d - e), with these factors proving very important for the BLR model classifications, the 2019 landslides occurred across the whole landscape, with no preferential occurrence at any aspects or slope angles (E.g. Fig. 8d - e). This, combined with the homogenous categorical input data, meant that there was very little spatial information for the BLR model to use when attempting to classify landslide occurrence and non-occurrence.


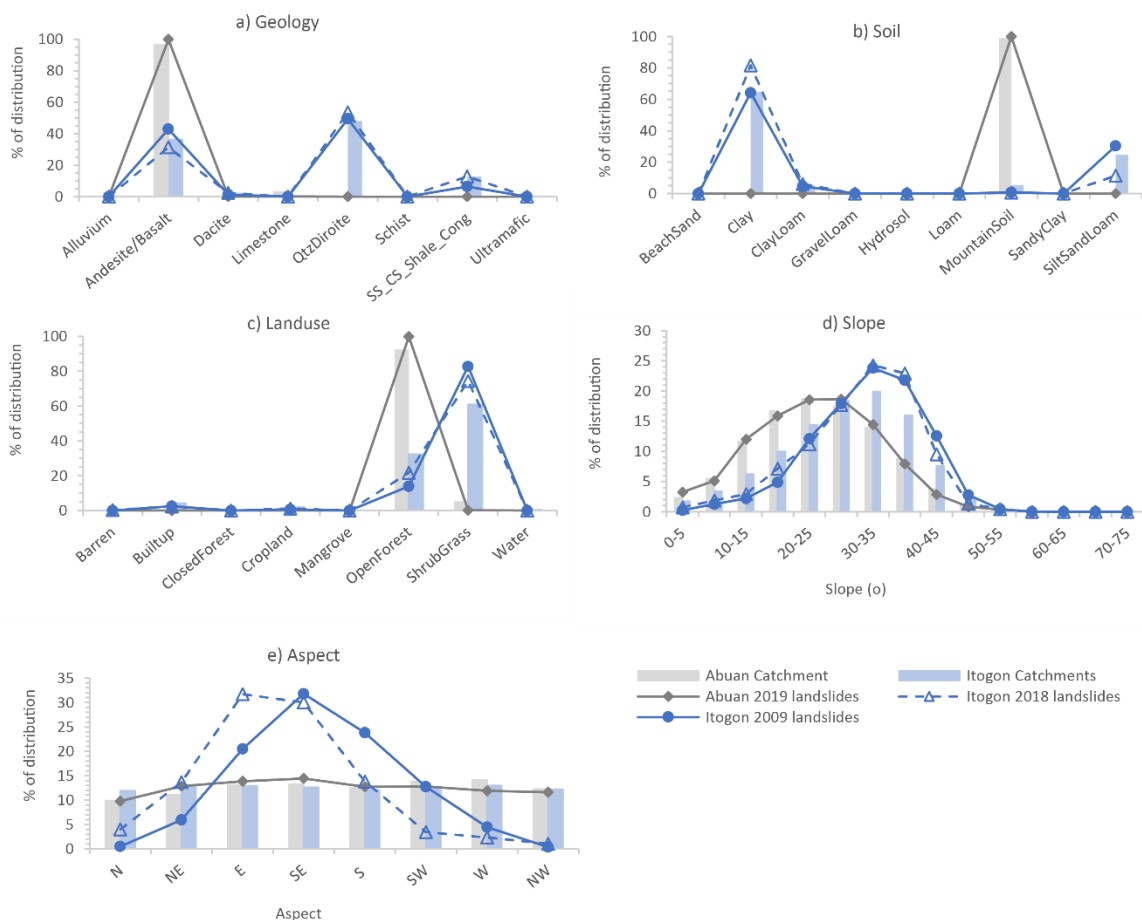

***Figure 8.*** *The distributions of each study region and of different landslide datasets with respect to a)*
*geology, b) soil, c) land use, d) hillslope angle, and e) hillslope aspect.*


However, whilst this may offer an explanation as to why the 2019 model was inaccurate, it raises the
question of why the landslides in 2019 were so evenly distributed across the landscape. One potential
explanation is based on the landslide storm cell model proposed by Crozier (2017). This model describes
how patterns of landsliding associated with atmospheric storms cells, whereby landslides are assumed to
occur in a circular pattern radiating from the storm centre. The model defines three main zones. The core
zone, where a storm has maximum rainfall totals and intensity (typically >500 mm of total rainfall) and
associated landslides that are distributed at all landscape locations regardless of variations in land use,





geology, and topography. The middle zone, with enough rainfall to trigger widespread landsliding, but only where the landscape is particularly susceptible to failure due to factors such as higher slope angles, and more

failure-prone lithologies, land-use, and soil cover. And the peripheral zone, where landslide occurrence is restricted by insufficient rainfall, and therefore only limited landsliding occurs in portions of the landscape where rainfall can accumulate. In this case, if the Abuan catchment occurred within the core zone of the Kammuri typhoon, then this could explain why the landslides occurred across all portions of the landscape. To test this, daily PERSIANN-CCS rainfall data for the 2019 Kammuri typhoon in Abuan and the 2018

Mangkhut and 2009 Parma typhoons in Itogon were obtained. These data show that the maximum daily rainfall across Abuan during the 2019 Kammuri typhoon was actually lower than that which occurred across Itogon during both the 2018 and 2009 typhoons (78 mm compared to 97 and 108 mm). This suggests that either the PERSIANN-CDR data are significantly underestimating the 2019 event rainfall, which such satellite-derived rainfall products are prone to doing (Zhu *et al.* 2016; Jiang *et al.* 2017), or that particularly

intense rainfall is not the cause of the observed landslide distributions. Given that the Abuan region was never in the centre of the path of the Kammuri typhoon (where you would expect the core zone to occur), the latter is potentially more likely.

Another possible explanation for the 2019 landslide distributions could be the directionality of the 2019

typhoon. It has been proposed that the direction of storm wind circulations influences the aspects at which landslides occur (e.g. Gorokhovich & Vustianiuk 2021). As such, if the 2019 Kammuri typhoon changed direction several times, and thus wind circulations changed orientation, then this could explain why landslides occurred at all aspects and slope angles. However, for several reasons this explanation also seems unlikely. First, rainfall analysis by Abancó *et al.* (2021) found that typhoon rain and wind direction could not

explain the aspect response of the 2018 Mangkhut landslides, suggesting that typhoon directionality is not as important a process as might be expected. Second, we could find no evidence that the 2019 typhoon changed direction, with it seemingly following a relatively constant east to west path.

As such, overall, it is suggested that the inaccuracies in the 2019 model are due to a combination of poor

input data and the fact that the 2019 Kammuri landslides were distributed evenly across the landscape (therefore giving little spatial correlation for the BLR method to use for classification). However, it remains unclear why the landslides triggered by the 2019 typhoon were distributed like this, with the rainfall in 2019 being no more intense than the other events, and no obviously strange directional behaviour of the Kammuri typhoon. Future work should investigate this issue further, with a particular focus on an analysis of the 2019





event rainfall, as understanding why certain typhoon events may trigger unpredictable landslides has important implications for the forecasting and managing of future landslide hazard.

The second question raised by the inaccuracies of the 2019 Abuan model, is a general point about whether it is better to have no susceptibility map or a poor susceptibility map? Instinct may be to assume a mantra of
"something is better than nothing". However, in reality, this is unlikely to be the case. Susceptibility maps are commonly used for important hazard management purposes. As such, inaccurate landslide susceptibility maps could lead to ineffective, inappropriate, or insufficient hazard management strategies being implemented. Not only is this likely to waste resources, but it could also present a danger to life and development. For example, if regions incorrectly classified as low susceptibility are subsequently built on,
then human and infrastructure vulnerability could be increased.

### 6.2 Landslide time dependency

As described in the aims, the final objective of this paper is to provide insight into the potential time-dependent nature of typhoon-triggered landslide susceptibility in the Philippines.  This objective essentially operates under the hypothesis that if typhoon triggered landslides are time dependent, then in a given region
there will be a reduced model accuracy when using a model developed from one typhoon event to classify another. In this case, we tested this by using AUROC cross-validation to compare the 2018, 2009 and 2018-2009 typhoon-triggered landslide susceptibility models. It should be noted that we do not consider 2019 Abuan model here as this event occurred in a different region, which would therefore introduce issues of spatial dependency, which are not the focus of this paper.


As outlined in the results, we see a 9 – 10% reduction in model accuracies when using the single event models from 2018 or 2009 to classify independent landslide data from the other typhoon event compared to the accuracies obtained when training/testing using each typhoon event separately (Figs. 5 and 7). This therefore supports out initial hypothesis, suggesting that, as observed in other regions and landslides (e.g.,
monsoon-triggered landslides in Nepal: Jones *et al.* 2021), there is some degree of time-dependency in the susceptibility of typhoon-triggered landslides. But what are the implications of this for landslide hazard management? It is well described that landslide susceptibility models are regularly used for purposes that involve the forecasting of future landslide events (Reichenbach *et al.* 2018; Palau *et al.* 2020). Consequently, in the Philippines, issues of time-dependency mean that a hazard manager cannot be confident that a
landslide susceptibility model developed from a single past typhoon event will actually be accurate and





reliable when forecasting landslides associated with a future typhoon. This is clearly a problem, as it fundamentally affects the use of susceptibility models for their primary (or at least common) application. So what are the solutions to this problem?

One solution could be to ensure that typhoon-triggered landslide susceptibility models are always developed using landslide data from multiple typhoon events. As shown in Fig. 6, the susceptibility model developed using the training landslide data from the 2018 and 2009 events combined had higher overall accuracies when classifying the independent data from each event separately. Indeed, the 2018+2009 model was only 1% less accurate at classifying the 2009 data than the 2009 model was, and only 4% less accurate at 595 classifying the 2018 data than the 2018 model was. Statistically, this result is logical, as, in effect, the combined 2018+2009 model coefficients will be an average of the optimum coefficients that would be obtained when modelling each event in isolation, and therefore the multi-event model has a more generalised ability to classify both events. In contrast, the single event model coefficients are very specific to one event, and therefore are notably less accurate when used to classify a different event. This suggests that whilst a 600 single event model will likely be more accurate at classifying the specific event used to train that model, a multi-event model will be more generalizable and thus appropriate for forecasting an unknown future typhoon event. Furthermore, it would be logical to assume that the more events combined in the training of a susceptibility model, the more generalizable that model will be. This idea is corroborated by several other studies. For example, Ozturk *et al.* (2021) find that susceptibility model accuracy increases with increasing 605 landslide inputs until a saturation point occurs when a large enough portion of a study region has observed a landslide and been included in the model training. Similarly, Jones *et al.* (2021) found that monsoon-triggered susceptibility model accuracy increased significantly as you moved from using a single monsoon season of data to using approximately 10 monsoon seasons of data, with a levelling off of the accuracy as you added more seasons of landslide data beyond this. In this case, it is therefore reasonable to assume that given 610 more typhoon triggered landslide data, the generalisability and overall forecasting reliability of the multi-event model could be improved. This of course requires robust testing in future work when more typhoon-triggered landslide data become available.

**7.0 Conclusions**

In conclusion, using BLR techniques we have developed new and updated susceptibility maps for the Itogon 615 and Abuan regions of the Philippines, with the 2009, 2018, and combined 2009-2018 models being considerably more accurate (78 – 82%) than the 2019 model (65%). We find that the three typhoon events




caused quite different landslide responses. Most notably, landslides in Itogon were heavily distributed across E/SE/S-facing slopes and at slope angles >30º, whereas landslides in Abuan occurred across all aspects and slope angles. The uniform distribution of 2019 landslides across all parts of the landscape combined with

homogenous input datasets for geology, soil and landcover is likely to be the cause of the lower accuracy of the 2019 model. Finally, the AUROC validation shows that using a susceptibility model for one typhoon event to forecast/hindcast another leads to a 6 – 10% reduction in model accuracy compared to the accuracy obtained when modelling and validating each event separately. This suggests that typhoon-triggered landslides in the Philippines display some degree of time-dependency. However, using a susceptibility model

for two combined typhoon events (2009 + 2018) to forecast/hindcast each typhoon event separately led to just a 1 – 3% reduction in model accuracy. This suggests that combined multi-event typhoon-triggered landslide susceptibility models will be more accurate and reliable for the forecasting of future typhoon-triggered landslides.

**Data and code availability**

The landslide inventory will be publicly available at the end of the SCaRP project (NE/S003371/1), currently scheduled for April 2022. It will be accessible at the NERC Environmental Information Data Centre (EIDC) deposit: https://eidc.ac.uk/ (Environmental Information Data Centre). No custom developed code is used in this research, all code used was derived directly from the existing glmnet package:
https://cran.r-project.org/web/packages/glmnet/glmnet.pdf

**Author contributions**

GB and FT conceived the project. MM and CA led the development of the landslide inventories. JJ led the data analysis, modelling and predisposing factor data collection, with input from GB, CA, and MM. JJ wrote up the manuscript, which was edited and reviewed by GB, FT, CA and MM.

**Competing interests**

All authors declare no competing interests.



## Acknowledgements

We would like to thank Rose Ann F. Amado and Engr. Paolo A. Antazo for their assistance with mapping the 2019 Kammuri inventory. We are also grateful to the Mines and Geosciences Bureau (MGB) for providing the hazard map shown in Fig 1 and the geological data for both Itogon and Abuan and to Xavier Fuentes for making the soil and land cover data available through https://www.geoportal.gov.ph/. The researchers based in the UK (Joshua Jones, Georgina Bennett and Clàudia Abancó,) have been funded by the NERC Newton Agham fund (contract NE/S003371/1, project SCaRP), and the researchers based in the Philippines (Mark Anthony Matera and Fibor Tan) have been funded by the Department of Science and Technology – Philippine Council for Industry, Energy, and Emerging Technology Research and Development (DOST-PCIEERD, project no. 07166).

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

**Figure and table captions**

**Figure 1.** Current landslide susceptibility map of the Itogon region (Mines and Geosciences Bureau, (BGB),
770   2018).

**Figure 2.** Locations of the two study regions, including the three landslide inventories used throughout this paper.

**Table 1.** Summary of the BLR-LASSO results obtained for each of the four developed models. All numbers are the mean of the 50-runs undertaken for each model. The AUROC value is the accuracy of the model at classifying the landslide data used to train the model.

**Figure 3.** Graphical display of the aspect regression coefficients presented in Table 1.

**Figure 4.** The final susceptibility maps produced from: a) the 2018 landslide inventory, b) the 2009 landslide inventory, c) the combined 2018 and 2009 inventories, and d) the 2019 inventory.

**Figure 5.** Receiver Operator Curves (ROCs) showing the success of a) the 2018 model classifying the
independent 2018 testing data, b) the 2009 model classifying the independent 2009 testing data, c) the 2019 model classifying the independent 2019 testing data, and d) the 2018+2009 model classifying the independent 2018+2009 testing data. In each case the reference ROC for a "random" model with 50% accuracy is shown, as is the mean Area Under the Receiver Operator Curve (AUROC) value for all ROCs.



**Figure 6.** Receiver Operator Curves (ROCs) showing the success of the 2018+2009 model at classifying independent testing data from a) 2018, b) 2009, c) 2019. In each case the reference ROC for a "random" model with 50% accuracy is shown, as is the mean Area Under the Receiver Operator Curve (AUROC) value for all ROCs.

**Figure 7.** Receiver Operator Curves (ROCs) showing the success of a) the 2018 model classifying independent testing data from 2009, b) the 2018 model classifying independent testing data from 2019, c) the 2009 model classifying independent testing data from 2018, and d) the 2009 model classifying independent testing data from 2019. In each case the reference ROC for a "random" model with 50% accuracy is shown, as is the mean Area Under the Receiver Operator Curve (AUROC) value for all ROCs.

**Figure 8.** The distributions of each study region and of different landslide datasets with respect to a) geology, b) soil, c) land use, d) hillslope angle, and e) hillslope aspect.