# Peer review of "Multi-event assessment of typhoon-triggered landslide susceptibility in the Philippines"

_Natural Hazards and Earth System Sciences, 2022_

## Author Response (AR1)

Note, line numbers correspond to the non-tracked change version of the manuscript.

RC 1

General Comments

General Comment 1: First, the results of two different study areas considering 3 events across time that aim to assess a research objective that operates on the hypothesis that the time dependence of typhoon-triggered landslides in a region would be evident in the deterioration of model accuracy.The results of the Itogon region with events from 2009 and 2018 sufficiently address this research question and provide quantified information on the temporal behavior of landslide susceptibility across time. The analysis of these results should already merit publication.

The inclusion of the results in the 2019 landslides from Abuan deviate from the direction of evaluating time-dependent susceptibility. The comparison of the model in Abuan to Itogon veers towards investigating regional and spatial differences between the sites in which these typhoon-triggered landslide occurred. I would recommend a separate study to focuses on the spatial and not temporal aspect of typhoon-triggered susceptibility be considered for the Abuan results.

*Response: We agree that a separate full study would be needed to properly assess issues of spatial dependency. However, we are of the opinion that inclusion of the Abuan inventory does add value to the paper. For example, it facilitates the discussion around whether a bad model is better than no model (which R2 was particularly keen to see highlighted more in the paper). It also allows us to set the scene for recommending the separate study as suggested by the reviewer.*

*Furthermore, we have updated the abstract (lines 23 – 35), the introduction and objectives (lines 80 – 100), and discussion (lines 641 – 655) to make it clear that the spatial issues are not the central part of the paper, and to generally link that part of the discussion more explicitly to the objectives/introduction.*

General Comment 2: Second, the authors could consider referencing an updated Landslide Hazard Atlas of susceptibility maps generated by the University of the Philippines Resilience Institute and the Nationwide Operational Assessment of Hazards (NOAH), available at https://noah.up.edu.ph, rather than the MGB susceptibility maps. The landslide hazard maps are available on a national level and are used in practice for hazard zonation and land use planning. A large section in their discussion could benefit from comparing their results to the NOAH hazard information.

*Response: We thank the reviewer for pointing out this hazard data base. We agree that referencing it will benefit the paper. As such, we briefly describe NOAH on lines 45 – 51, provide an example map from the NOAH database in Figure 1 (line 60), and to the discussion/comparison on lines 467 – 473.*

General Comment 3: The most important contribution of this study to the community is the quantified deterioration of susceptibility model performance accuracy in Itogon that typhoon-triggered landslides display a degree of dependency across time. Overall, I recommend that the authors update their hazard information for better context in the discussion and to highlight the improvement of susceptibility information with multi-temporal landslide inventory. I also recommend that the authors contemplate on the exclusion of the 2019 Abuan landslides in this study. The results do not support the research objective's underlying hypothesis to consider the time-dependence of typhoon-triggered landslides.

*Response: As outlined in the previous two responses, we have added reference to the NOAH maps where relevant, and whilst we have decided to keep in the Abuan inventory, we have made it clear why, and added text to the introduction/objectives and discussion to make it clear that the spatial dependency is not the main focus of the paper.*

Specific Comments

L45-50: Consider incorporating the landslide hazard information from the NOAH Landslide Hazard atlas. This information would be beneficial to further realizing the contribution made by this study in Itogon for typhoon-triggered landslides.

M.L. Rabonza, R.P. Felix, A.M.F Lagmay, R.N. Eco, I.J. Ortiz, ang D.K. Aquino (2015). Shallow landslide susceptibility mapping using high-resolution topography for areas devastated by super typhoon Haiyan. Landslides, Volume 13, Issue 1 pp 201-210

Alejandrino, A.M.F. Lagmay and R.N. Eco (2016) Shallow Landslide Hazard Mapping for Davao Oriental, Philippines Using a Deterministic GIS ,Model. In: Communicating Climate Change and Natural Hazard Risk and Cultivating Resilience: Case Studies for a Multidisciplinary Approach Eds. Yekaterina Y. Kontar. Springer, Berlin Germany

Paul Kenneth Luzon, Kristina Montalbo, Jam Galang, Jasmine May Sabado, Carmille Marie Escape, Raquel Felix, and Alfredo Mahar Francisco Lagmay (2016) Hazard mapping related to structurally controlled landslides in Southern Leyte, Philippines. Natural Hazards and Earth System Sciences, 16, 875-883, 2016

*Response: We agree this is useful. Have added reference to this on lines 45 – 51, provide an example map from the NOAH database in Figure 1 (line 60), and to the discussion/comparison on lines 467 – 473.*

L61-63: The concept of spatial and temporal dependence introduced in this section could be strengthened by a connection to the path-dependence of landslides by Temme et al. (2020).

Temme, A., Guzzetti, F., Samia, J., & Mirus, B. B. (2020). The future of landslides' past—A framework for assessing consecutive landsliding systems. Landslides, 17(7), 1519–1528. https://doi.org/10.1007/s10346-020-01405-7

*Response: We agree. Have described the concept and added reference to this on lines 66 – 70.*

L93: The use of the term time-dependence could pertain susceptibility during typhoon season, or within a sub-seasonal period. I recommend the authors to consider rephrasing this to a path-dependent perspective and connect to the concepts of Temme et al. (2020) and the results of the multi-temporal susceptibility analysis of Samia et al. (2020).

 Samia, J., Temme, A., Bregt, A., Wallinga, J., Guzzetti, F., & Ardizzone, F. (2020). Dynamic path-dependent landslide susceptibility modelling. Natural Hazards and Earth System Sciences, 20(1), 271–285. https://doi.org/10.5194/nhess-20-271-2020

*Response: We have added clarity to the term time dependence on lines 97 - 98, but do not wish to use the exact phrasing of Samia et al. in this context. This is because the Samia path dependent terminology is used specifically for cases where the locations of previous landslides influence future landslides. However, time-dependence is a broader term that could include other causes of dependency such as earthquake preconditioning or other dynamic landslide changes.*

L189: Why was the inventory slightly clipped? It also would be worth mentioning a brief qualitative comparison between this inventory 2018 Mangkhut and that of Emberson et al. (2022).

Emberson, R., Kirschbaum, D. B., Amatya, P., Tanyas, H., & Marc, O. (2022). Insights from the topographic characteristics of a large global catalog of rainfall-induced

landslide event inventories. Natural Hazards and Earth System Sciences, 22(3), 1129–1149. https://doi.org/10.5194/nhess-22-1129-2022

**Response: The reason is to maintain the catchment boundaries (the initial inventory spread into other catchments. This is added in brackets on lines 195 – 196. Reference to Emberson paper added on lines 222 – 224 (where we felt this comparison fitted better).**

L371-400: These paragraphs are presented in a way that focuses on events across time, but gives the impression that the 2009, 2018 and 2019 landslides occurred on spatially similar settings or even same site. Splitting the presentation of results into two paragraphs (one for Abuan and one for Itogon) to discuss the separate geographic sites could make it clearer.

**Response: This section has been re-structured as suggested to keep Itogon and Abuan separate on lines 369 – 414.**

L455-473: Is there any insight on the hazard between 2009 and 2018 in Itogon that can be derived from the susceptibility models? Any insight on susceptibility or changes that could've caused landslides to occurred with smaller passing tropical cyclones within these 9 years? (Referring as well to insight from Figure 4)

**Response: It is hard to make any comment on passing cyclones within that period as we lack landslide data for those events, so can't really quantify how the susceptibility changes in-between the two time periods (i.e. we can just compare the start and end of the period). This would require landslide data from more time periods in between 2009 and 2018. This is a good point to raise though, so we add reference to passing storms and their potential impacts on path dependency to lines 610 – 615.**

L474-565: Please refer to the Landslide Hazard information from the susceptibility maps of NOAH to provide an updated hazard context for this section of the discussion.

**Response: We now refer to his on lines 469 – 474.**

L549-556: These are valid concerns and points of uncertainty raised for the Abuan susceptibility results. Though, the alignment of these results the objectives presented in L93-95 are not clear.

***Response: We have updated the abstract (lines 23 – 35), the introduction and objectives (lines 80 – 100), and discussion (lines 641 – 655) to make it clear that the spatial issues are not the central part of the paper, and to generally link that part of the discussion more explicitly to the objectives/introduction.***

L529-538: While magnitude underestimation is a limitation in the use of satellite-derived rainfall products, another factor worth discussing is the limitation to capture spatial patterns and locate the storm centers when using such products. (See Ozturk et al., 2021)

Ozturk, U., Saito, H., Matsushi, Y., Crisologo, I., & Schwanghart, W. (2021). Can global rainfall estimates (satellite and reanalysis) aid landslide hindcasting? Landslides, 18(9), 3119–3133. https://doi.org/10.1007/s10346-021-01689-3

***Response: We agree that this is an interesting addition to the discussion, so we have added a sentence to mention/reference this as suggested on lines 543 – 545.***

L590-612: Table 1. Shows that land cover is significant for the 2009 and combined 2009+2018 model. It would worth mentioning the role of land cover change that could have an influence susceptibility over time. Itogon is estimated have had significant tree cover loss between 2010 and 2020 based on: Global Forest Watch, http://globalforestwatch.org.

C. Hansen, P. V. Potapov, R. Moore, M. Hancher, S. A. Turubanova, A. Tyukavina, D.Thau, S. V. Stehman, S. J. Goetz, T. R. Loveland, A. Kommareddy, A. Egorov, L. Chini, C. O. Justice, J. R. G. Townshend, High-resolution global maps of 21st-century forest cover change. Science 342, 850–853 (2013).

***Response. This is a nice point, we have added some sentences to discuss this on lines 597 – 604.***

Technical Corrections

L32: '>30o' to >30°

***Response: This line has now been removed from the abstract.***

Figure 6. Consider using 'performance' rather than 'success'

Figure 7. Consider using 'performance' rather than 'success'

***Response: Changed on lines 445, 451, 855, and 860.***

RC2

Two general comments:

1. The discussion about the accuracy of the Abuan model includes discussion on whether "something is better than nothing" with regard to susceptibility mapping. The reviewer thinks this is a very relevant comment, and should be highlighted in the conclusions of the manuscript. If the authors believe the input data into the model is not sufficient enough to produce a reliable susceptibility map, it should not be concluded that a new susceptibility map is produced, particularly when the region did not already have a map available.

***Response. We agree with the reviewer and have updated our discussion accordingly in a new section (6.1.1; lines 565 – 576).***

2. The aim of the paper was to use data from multiple typhoon events to assess typhoon-triggered landslide susceptibility in the Philippines. The reviewer thinks the topic of time-dependance is discussed sufficiently within the Itogon region, with analysis completed on two individual typhoon events and then a combination of the two events. These three models are then tested on the 2019 data from the Abuan region, with poor results (AUROC between 0,54 and 0,59 according to Figure 6 and 7). To the reviewer this seems that some discussion is warranted on the spatial dependancy, although it is mentioned in Line 574 that this is not the focus of the paper (but not mentioned or excluded from the paper in the introduction or abstract). If the focus of the paper is really only discussing time dependence, it may not be relevant to include the Abuan region, which is only analysed using one typhoon event.

*Response. We have added some explicit discission about spatial dependency in a new section on lines 641 – 652. We have also updated the abstract (lines 23 – 35), the introduction and objectives (lines 80 – 100) to make it clear that the spatial issues are not the central part of the paper, and to generally link that part of the discussion more explicitly to the objectives/introduction.*

Specific comments:

- Section 2 general comment - There is a mixing of unit systems here. Amount of rain is listed in millimeters, while wind speeds are noted in mils per hour.

*Response: We have amended all units to be SI. I.e. All mph changed to kph. Lines 134-135, 153, 168.*

-The accuracy of the model from the 2009 typhoon was classified as "good/excellent", and the combined 2009/2018 model as "good", are there any complications with building a susceptibility model from a typhoon event which was described in Section 2.1 as influenced by the Fijiwhara effect, where the typhoon was impacted/worsened by a nearby typhoon?

*Response. We think this is an interesting question. It is hard to answer without detailed rainfall data, but we have add a sentence or two to mention the issue in the discussion and how it could have had an impact on the susceptibility modelling on lines 594 – 597.*

- Line 227 - the reviewer thinks the toolbox in ArcGIS may be called "Spatial analyst", not Spatial Analysis.

*Response. The reviewer is correct, we have amended the text on line 238.*

- Line 286-289 - In point 4 it would be nice to mention what the other predisposing factors are. Here it is listed that three factors are categorical, but one must look to Table 1 to find the other factors.

*Response. We now list all factors on lines 297 – 300.*

- Line 443-447 - The figure caption for Figure 7 was challenging to read, with similar years being discussed. Perhaps make it more clear on the figures that a and b are using a different model year than c and d.

**Response. We have now added legend elements to the figure (next to the subpanel a,b,c labels) to describe what each model shows on the figure. See line 450.**

- Line 449 - the word "models" is missing after 2009.

**Response. Missing word added on line 456.**

- Figure 8 - In the plot for e) Aspect, the reviewer does not understand why the distributions for the Itogon catchments have a peak at E/SE aspects, when the bar charts are approximately equal to the Abuan catchment data.

**Response. This is because in the Itogon case, landslides were far more likely to occur at these aspects. Whereas in Abuan, landslides occurred at all aspects. I.e. the distributions of aspects across the landscape are similar in both regions, but landslides are preferentially occurring at E/SE aspects in Itogon, but not Abuan. We clarify this with an explanation added to figure caption on lines 517 – 519.**

- Line 520-527 - The sentences discussing the three main zones (core zone, middle zone and peripheral zone) are not really sentences and are slightly challenging to read. Consider restructuring.

**Response. We have now edited these sentences, so they are easier to read, on lines 525 – 534.**